# Association between Three Low-Carbohydrate Diet Scores and Lipid Metabolism among Chinese Adults

**DOI:** 10.3390/nu12051307

**Published:** 2020-05-03

**Authors:** Li-Juan Tan, Seong-Ah Kim, Sangah Shin

**Affiliations:** Department of Food and Nutrition, Chung-Ang University, Gyeonggi-do 17546, Korea; tanlijuan88@cau.ac.kr (L.-J.T.); sakim8864@gmail.com (S.-A.K.)

**Keywords:** LCD score, CHNS, dyslipidemia, dietary factor, plant based, animal based, Chinese adults

## Abstract

This study investigated the blood lipid levels of 5921 Chinese adults aged >18 years using data from the China Health and Nutrition Survey 2009. Diet information was collected through 3 day, 24 h recalls by trained professionals. The low-carbohydrate diet (LCD) score was determined according to the percentage of energy obtained from carbohydrate, protein, and fat consumption. Dyslipidemia was defined when one or more of the following abnormal lipid levels were observed: high cholesterol levels, high triglyceride levels, and low high-density lipoprotein cholesterol levels. Multivariate adjusted odds ratios (ORs) and their 95% confidence intervals (95% CIs) were calculated using logistic regression models. After adjusting the confounding variables, in males, the OR of hypercholesterolemia was 1.87 (95% CI, 1.23–2.85; *p* for trend = 0.0017) and the OR of hypertriglyceridemia was 1.47 (95% CI, 1.04–2.06; *p* for trend = 0.0336), on comparing the highest and lowest quartiles of the LCD score. The animal-based LCD score showed a similar trend. The OR of hypercholesterolemia was 2.15 (95% CI, 1.41–3.29; *p* for trend = 0.0006) and the OR of hypertriglyceridemia was 1.51 (95% CI, 1.09–2.10; *p* for trend = 0.0156). However, there was no significant difference between plant-based LCD scores and dyslipidemia. In females, lipid profiles did not differ much among the quartiles of LCD scores—only the animal-based LCD score was statistically significant with hypercholesterolemia. The OR of hypercholesterolemia was 1.64 (95% CI, 1.06–2.55), on comparing the highest and lowest quartiles of the LCD score. In conclusion, a higher LCD score, indicating lower carbohydrate intake and higher fat intake, especially animal-based fat, was significantly associated with higher odds of hypercholesterolemia and hypertriglyceridemia in Chinese males. Future studies investigating the potential mechanisms by which macronutrient types and sex hormones affect lipid metabolism are required.

## 1. Introduction

With an increasing prevalence worldwide, dyslipidemia has become a public health problem. Dyslipidemia is a well-known important risk factor for cardiovascular disease (CVD) and metabolic syndrome [1]. In 2012, the mortality rate of chronic non-communicable diseases in China was 533 per 100,000, including 272 deaths caused by cardiovascular and cerebrovascular diseases [2]. In the past 30 years, with rapid economic development and improvement in the standard of living in China, the blood lipid levels in the Chinese population have gradually increased, and the prevalence of dyslipidemia has increased significantly. In 2012, the prevalence of hypercholesterolemia was 4.9%, which is 69% higher than in 2002 (2.9%); the overall prevalence of dyslipidemia in Chinese adults was 40.4% in 2012, which was significantly higher than that in 2002 (18.6%) [2]. This indicates that the health care burden of dyslipidemia and its related diseases has consistently increased in the past decade.

Dyslipidemia is a disease characterized by elevated total cholesterol (TC) and/or triglyceride (TG) levels or a low high-density lipoprotein cholesterol (HDL-C) level, which contributes to the development of coronary heart disease and stroke [1]. According to the data reported by the Chinese Center for Disease Control and Prevention in 2015, 26.4% of ischemic heart disease cases in China are attributed to hypercholesterolemia, which is one component of dyslipidemia. Secondary dyslipidemia accounts for most dyslipidemia cases and is associated with lifestyle factors or medical conditions that interfere with blood lipid levels over time. Among these influencing factors, obesity (specifically abdominal obesity), alcohol consumption, and high-fat diets (specifically, high saturated and trans-fatty acid intakes) are well-known risk factors of dyslipidemia, as confirmed by several previous studies [3,4,5].

Carbohydrates significantly contribute to the total energy intake, particularly in the Chinese population who use rice and wheat as their staple food. At present, several studies have assessed the effect of carbohydrate intake on dyslipidemia [6,7,8,9,10]. Moreover, most of these studies have been conducted in European and American populations, and a few Asian studies have been conducted in the Korean population [9,10]. A study assessing the effect of carbohydrate intake on dyslipidemia in the Chinese native population has not been conducted yet.

The concept of and calculations to determine the low-carbohydrate diet (LCD) score have been introduced in detail previously [9,11,12]. In summary, participants with the highest scores have the lowest carbohydrate intakes. In 1992, the percentage of energy intake from carbohydrates in urban and rural residents in China was 66.2%, which decreased to 55.0% in 2012 according to The Report on the Status of Nutrition and Chronic Diseases of Chinese residents (2015). Therefore, it is of epidemiological value to study the effect of carbohydrate intake on dyslipidemia in the Chinese population. The present study used the low-carbohydrate diet (LCD) score to examine the association between LCD and dyslipidemia and its components in Chinese adults. Further, we identified the association by source foods (plant based and animal based).

## 2. Materials and Methods

### 2.1. Study Population

Data were obtained from the China Health and Nutrition Survey (CHNS), which used a multistage, random cluster sampling process to select samples from 15 provinces in China [13]. The sample scheme is reported in detail elsewhere [13]. Data on fasting blood glucose levels were first collected in 2009 (N = 9549). In the CHNS, the information was collected by trained interviewers to assure compliance and quality of data. The CHNS dataset and the Institutional Review Board information are available on the official website of the CHNS (https://www.cpc.unc.edu/projects/china). Participants provided written informed consent for inclusion in this study.

This study used one round of survey data from 2009. Participants aged less than 18 years (*n* = 1160) were excluded. Subsequently, we excluded 54 participants with insufficient blood sample data (i.e., TC, HDL-C, low-density lipoprotein cholesterol [LDL-C], TG, and fasting plasma glucose levels) and 22 participants with implausible energy intake (<800 kcal/day or >6000 kcal/day for males; <600 kcal/day or >4000 kcal/day for females) [14]. Additionally, we excluded participants taking medications for hypertension, diabetes, and myocardial infarction and participants who reported a medical history of heart attack, stroke, or diabetes (*n* = 2392). Finally, a total of 5921 participants (2743 males and 3178 females) were included in this study. A flowchart of the selection process of the study population is shown in Figure 1.

### 2.2. Dietary Assessment and Calculation of the Low-Carbohydrate Diet Score

The 3 day, 24 h dietary recalls were used to assess the dietary intake of each participant. The quantities and types of food and beverages consumed during the past 24 h were determined from the participants. The interviewers measured the participants’ dietary intake with picture aids and food models during household interviews. The participants’ energy and nutrient intakes were calculated using the China Food Composition Table (FCT) (FCT 2002 edition and FCT 2004 edition).

To evaluate LCD based on the method used by Halton et al. [11], we referred to the concept of the LCD score. According to the percentage of energy from carbohydrate, protein, and fat consumption, the participants were ranked separately and were subsequently divided into 11 groups. Using the assignment method by Halton et al., the highest score (LCD score = 30) group had the lowest carbohydrate intake and the highest fat and protein intakes, while the lowest score (LCD score = 0) group had the highest carbohydrate intake and the lowest fat and protein intakes. To facilitate the analysis, we further calculated the scores for animal-based food (the percentage of energy from carbohydrate, animal protein, and animal fat consumption) and plant-based food (the percentage of energy from carbohydrate, plant protein, and plant fat consumption) in a similar manner. The division of animal- and plant-based food is based on the FCT 2002 and FCT 2004 editions. Table 1 presents the percentages used to determine the LCD score, animal-based LCD score, and plant-based LCD score. According to the scores (from low to high), the participants were divided into four groups according to gender.

Simultaneously, based on the study by Kim et al. [9], when comparing the acceptable macronutrient distribution range (AMDR) for carbohydrates as recommended by the Chinese Dietary Reference Intakes (CDRIs) Handbook (2013), the participants in each quartile were divided into three groups according to the percentage of energy from carbohydrates: lower (<50%), standard (50–65%), and higher (>65%). Moreover, we adjusted the carbohydrate intake criteria according to previous research criteria [15,16]: low-carbohydrate diet (<40%), moderate-carbohydrate diet (40–65%), and high-carbohydrate diet (>65%). Finally, fat intake in each quartile was also determined in our study on comparing the AMDR using the fat intake criteria from the CDRIs Handbook: lower (<20%), standard (20–30%), and higher (>30%).

### 2.3. Determination of Dyslipidemia

Dyslipidemia is diagnosed when one or more of the following abnormal lipid levels are observed: high cholesterol levels (hypercholesterolemia), high triglyceride levels (hypertriglyceridemia), and low HDL-C levels based on the Guidelines for the Prevention and Treatment of Dyslipidemia in China (2007) [2]. Hypercholesterolemia is a disease characterized by increased TC levels in fasting serum (>240 mg/dL (6.22 mmol/L)) or increased LDL-C levels (>160 mg/dL (4.10 mmol/L). Hypertriglyceridemia is a disease characterized by increased TG levels in fasting serum (200 mg/dL (2.26 mmol/L)), and low HDL-C levels is a disease characterized by decreased HDL-C levels (<40 mg/dL (1.04 mmol/L)) [2].

### 2.4. Assessment of Other Variables

Body mass index (BMI) was obtained by dividing the body weight by the square of the height (kg/m^2^). Participants were divided into four groups based on their BMI values: underweight (BMI < 18.5 kg/m^2^), normal weight (≥18.5 kg/m^2^ and <24 kg/m^2^), overweight (≥24 kg/m^2^ and <28 kg/m^2^), and obese (BMI ≥ 28 kg/m^2^) according to the Guidelines for the Prevention and Control of Overweight and Obesity in Chinese Adults (2003). Sociodemographic variables such as age, gender, and income level and lifestyle variables such as alcohol consumption, smoking status, and physical activity level (PAL) were obtained using well-validated questionnaires distributed by trained interviewers. Information regarding alcohol consumption was divided into two groups according to frequency: “yes” (consumed alcohol more than once in the past year) or “no.” Information regarding smoking status was divided into three groups: “current smoker” (still smoking), “past smoker” (previously smoking but currently not smoking), or “never smoker”.

PAL included the following three domains: transportation activities, occupational activities, and sports. PAL was measured in terms of metabolic equivalent of task (MET) minutes in each week [17,18], which were converted using the time spent on each activity. The MET scores for each activity were based on the 2011 Compendium of Physical Activities. Physical activity was also divided into three categories: light PAL (<600 MET minutes/wk), moderate PAL (≥600 MET minutes/wk and <3000 MET minutes/wk), and vigorous PAL (≥3000 MET minutes/wk) [18,19].

The area of residence was categorized into “rural” and “urban”. According to the results of the questionnaire, “urban neighborhood” (primary sampling units = 36) and “suburban neighborhoods” (primary sampling units = 36) were classified as “urban”; “towns” (primary sampling units) and “rural villages” (primary sampling units = 108) were classified as “rural”. Other covariates that were defined in the model included age (years), educational level (less than primary school, middle school, technical school, college, or university or above), and ethnicity (Han, other ethnicities).

### 2.5. Statistical Analyses

Participants’ general characteristics, such as age, income level, BMI, obesity, alcohol consumption, smoking status, PAL, and nutrient intake were analyzed using the generalized linear model and chi-square test for continuous and categorical variables, respectively, according to the quartiles of LCD scores based on gender. Moreover, all results for the continuous variables are presented as the mean ± standard error, and the results for the categorical variables are presented as *n* (%). After adjusting for potential confounding variables (including age, educational level, body mass index, ethnicity, physical activity level, alcohol consumption, smoking status, and individual income), logistic regression models were used to calculate the odds ratios (ORs) of dyslipidemia and their 95% confidence intervals (95% CIs). All statistical analyses were performed using the Statistical Analysis System software version 9.4. A *p* value < 0.05 was considered statistically significant.

## 3. Results

Table 2 shows the general characteristics of the participants according to LCD score quartiles based on gender. The male and female LCD median scores both ranged from 5.00 in the lowest quartile (Q1) to 25.00 in the highest quartile (Q4). Moreover, participants with higher LCD scores had higher income levels, lived in urban areas, consumed alcohol, performed more physical activity, and had higher educational levels compared to participants in Q1 (all *p* < 0.05). Males in Q4 were more likely to have obesity (*p* < 0.05). Further, regarding lipid profiles, participants with higher LCD scores had higher levels of TC and LDL-C. Furthermore, males in Q4 had higher levels of TG than males in Q1 (all *p* < 0.05).

Macronutrient intake was assessed according to LCD score quartiles based on gender (Table 3). Carbohydrate intake was decreased and protein and fat intakes were increased according to the LCD score. Interestingly, intake of plant-based protein was decreasing according to the LCD score.

In Table 4, carbohydrate and fat intakes are presented according to the quartiles of the LCD score. Compared with the AMDR for carbohydrates as recommended by the CDRIs Handbook, only 34.85% of males and 36.25% of females consumed carbohydrates within the recommended level, whereas 58.95% of males and 58.34% of females consumed carbohydrates above the recommended level. None of the participants consumed carbohydrates below the recommended level in Q1. In total, 56.58% and 57.49% of males and females, respectively, had lower fat intake than the AMDR, and most of these participants were assigned to Q1 and Q2. According to previous research criteria, 39.7% of males and 40.59% of females consumed moderate-carbohydrate diets, whereas only 1.35% of males and 1.07% of females adhered to a LCD. All participants in Q1 consumed high-carbohydrate diets. Even in Q4, 94.88% of males and 95.87% of females consumed moderate-carbohydrate diets, whereas only 5.12% of males and 4.13% of females adhered to a LCD.

The multivariate adjusted ORs and their 95% CIs for the components of dyslipidemia according to the LCD score quartiles based on gender are summarized in Table 5. Male participants with higher LCD scores had a higher risk of hypercholesterolemia (OR, 1.87; 95% CI, 1.23–2.85) than those in Q1 (*p* for trend = 0.0017) after adjusting for age, educational level, BMI, ethnicity, PAL, alcohol consumption, smoking status, and income level. Meanwhile, a similar significant association between the LCD scores and hypertriglyceridemia was observed in males (OR, 1.47; 95% CI, 1.04–2.06) on comparison with participants in Q1 (*p* for trend = 0.0336) after adjusting for the mentioned confounding variables previously. However, in females, there were no significant associations between the ORs for dyslipidemia and LCD scores in comparison to the participants in Q1.

Table 6 illustrates the multivariate adjusted ORs and their 95% CIs for the components of dyslipidemia according to the animal- and plant-based LCD score quartiles based on gender. Comparing the fourth and the first animal-based LCD score quartiles, the multivariate ORs for hypercholesterolemia were 2.15 (95% CI, 1.41–3.29; *p* for trend = 0.0006) and the multivariate ORs for hypertriglyceridemia were 1.51 (95% CI, 1.09–2.10; *p* for trend = 0.0156) in males. In females, there was a borderline linear trend for hypercholesterolemia from the first quartile to the fourth quartile (*p* for trend = 0.0651). Regarding the plant-based LCD score, there was no significant linear trend for dyslipidemia from the first quartile to the fourth quartile both in males and females.

## 4. Discussion

Using the CHNS 2009, the present study examined the impact of dietary carbohydrate intake on dyslipidemia and its components using the LCD scores. It was found that higher LCD scores were significantly associated with higher intake of legumes, nuts, fish, and other non-carbohydrate food sources, consistent with the results of an improved carbohydrate quality index from an original study [20] (Table A1). Furthermore, in the present study, it was also found that participants with higher LCD scores lived in urban areas, had high income, consumed alcohol, performed light physical activity, and had higher educational levels. Among these, the PAL and drinking status have been widely studied and confirmed as important factors on lipid metabolism [4,21,22]. Due to their inescapable impact, in the present study, we adjusted PAL and drinking status as covariables in the multivariate model to minimize their impact, improve the validity of research results and focus on the analyses of the impact of diet on lipid metabolism. In the follow-up study, we will consider PAL and drinking status as exposure factors in order to study their impact on lipid metabolism.

Furthermore, it was found that a higher LCD score was significantly associated with a higher risk of hypercholesterolemia and hypertriglyceridemia in males after adjusting for eight confounding variables. These observations are consistent with the observations noted in a series of studies comprising Chinese adults [1,14]. Furthermore, higher animal-based LCD scores were also associated with a higher risk of hypercholesterolemia and hypertriglyceridemia in males and with a borderline increase in TC levels in females, while the plant-based LCD score was statistically insignificant in males and females.

We grouped the participants according to the quartiles of the LCD scores and there was a clear dose–response relationship between dietary macronutrients and the components of dyslipidemia, specifically for carbohydrates. In our study, the higher the LCD score, the lower the carbohydrate intake, the higher the fat intake, and the higher the risk of hypercholesterolemia and hypertriglyceridemia in males.

A previous study [14] reported that males with the highest carbohydrate intake (>68.2% of energy) had a significantly lower TC level after adjusting for several confounding variables. This result was consistent with the result of the present study, in which participants with high LCD scores had a higher risk of hypercholesterolemia than participants with lower LCD scores.

The effect of the LCD score on lipid metabolism can be analyzed from the perspective of a low-carbohydrate and high-fat diet using our results in Table 4. According to Volek et al. [16], carbohydrates are considered as the main source of energy and glucose in the general diet, and they directly or indirectly regulate the distribution of excessive dietary nutrients through insulin to regulate lipolysis and lipoprotein assembly and processing. Thus, carbohydrates affect the association between dietary intake of saturated fat and circulating lipid levels.

Furthermore, the group with high LCD scores obtained a higher percentage of energy from fat, as shown in Table 4. Moreover, 41% and 37.08% of male and female participants in Q4 had a high-fat diet, respectively, resulting in an increase in the serum TC and LDL-C levels [23,24]. Hence, the higher the LCD score, the higher the odds of hypercholesterolemia.

Recent studies have focused not only on the quantities of fat and carbohydrate intake but also on the quality of fat and carbohydrates. In our study, participants with higher LCD scores consumed less cereals and tubers, which are the main sources of dietary carbohydrates. Nuts contain high levels of unsaturated fatty acids [25], and various animal products have different saturated, polyunsaturated, and trans-fatty acid levels [16,26]. Limiting the intake of certain forms of carbohydrates is the preferred dietary strategy for improving cardiovascular health. Dietary fiber, which is a form of carbohydrate, reduces the risk of atherosclerosis and CVD [27]. Several types of fatty acids and their intake also have different effects on lipid metabolism [28,29]. Moreover, according to Acosta-Navarro et al. [30], higher consumption of an animal-based diet leads to higher TC levels. Hence, animal-based LCD scores were associated with differences in TC levels according to gender in this study. In both males and females, higher animal-based LCD scores were significantly associated with a higher risk of hypercholesterolemia, while the plant-based LCD score showed no statistical significance. This result is inconsistent with the result of a previous study [31], mainly because the previous study, which assessed plant-based diets, emphasized not eating meat or eating less meat. However, in our study, the plant-based LCD score refers to the percentages of protein and fat that the participants consumed from plant-sourced foods during the 3 day diet survey, and the participants themselves were not prohibited from eating meat (Table 1).

The difference between males and females in statistical significance according to our results is possibly attributed to the different body fat distributions and hormonal differences between males and females. In normal-weight individuals, significant differences are observed in lipid and fat distributions between males and females [32]. Serum TG levels are relatively lower and HDL-C levels are higher in females than in males (Table 2), due to the differences in gender hormones and body fat distribution [33,34]. Additionally, the response to a high-fat diet in terms of lipoproteins has a gender dimorphism [35]. These findings indicate that the effect of a high-fat diet on serum HDL-C levels is more significant in females than in males, as confirmed in our study, and a multivariate adjusted OR for HDL-C levels was also observed in our study (Table 5, OR, 0.48; 95% CI, 0.28–0.82).

This study based on the CHNS indicates that higher LCD scores result in a higher risk of hypercholesterolemia and hypertriglyceridemia in males, but we cannot exactly determine the amount and type of carbohydrate or fat intake using LCD scores. Therefore, the comprehensive effect of several macronutrients in the general diet on lipid metabolism should be further considered.

Although the data collection of CHNS 2009 is a little dated, the CHNS is a study that constitutes a wide age range and large sample size after adjusting for a comprehensive range of potential confounding variables. The analysis results of the representative datasets are reliable and can be extended to the general Chinese population. To the best of our knowledge, this is the first study to use LCD scores to study the risk of dyslipidemia among Chinese adults. Compared to a food frequency questionnaire, a 24 h recall is more accurate and has a lower estimation bias [13].

However, this study has several limitations. First, since this is a cross-sectional study, we were not able to determine the causal association between dietary macronutrients and the risk of dyslipidemia. Second, the dataset assessed dietary intake using 3 day, 24 h recalls, which may have a relatively limited correction for the internal differences among the participants and may be affected by seasons. Third, as we assessed confounding variables, such as PAL, using a questionnaire, bias may exist. Finally, although we adjusted for confounding variables, there are still unknown variables that can influence the results. Hence, larger-scale prospective studies are required to determine the association between LCD and dyslipidemia.

## 5. Conclusions

In conclusion, among Chinese males, the group with higher LCD scores has a higher fat intake (especially animal-based fat), a diet pattern that is closer to a LCD, and greater odds of hypercholesterolemia and hypertriglyceridemia. We advise that people should pay more attention to the proportion of macronutrients in their diet, especially animal-based fat. The association between LCD score and dyslipidemia is significantly complex. In the future, careful attention should be paid to the quality of macronutrients for preventing dyslipidemia in the Chinese population. Further prospective studies and those involving detailed dietary nutrient intake evaluations are required to verify these findings in Chinese and other populations.

## Figures and Tables

**Figure 1 nutrients-12-01307-f001:**
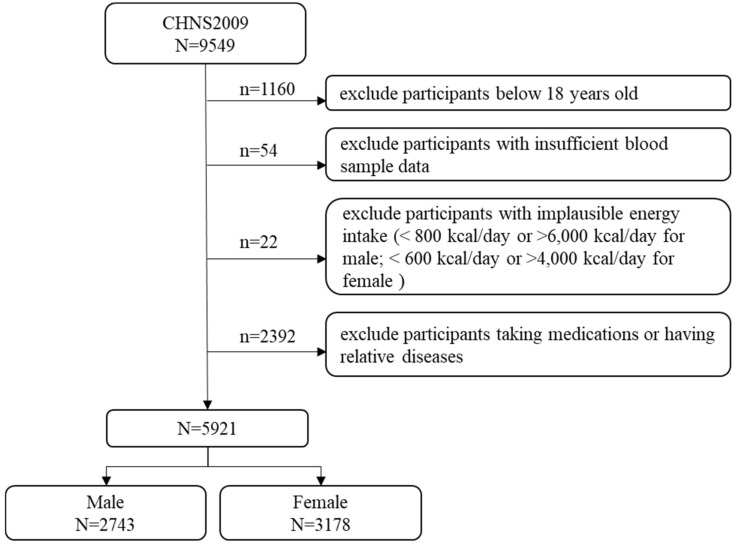
Selection process for the study population in the 2009 China Health and Nutrition Survey (CHNS).

**Table 1 nutrients-12-01307-t001:** Energy percent of macronutrients used in calculating the low-carbohydrate diet (LCD) scores, animal-based LCD scores, and plant-based LCD scores of Chinese adults, China Health and Nutrition Survey.

	Macronutrient Intake						
	Total Carbohydrate	Total Protein	Total Fat	Animal-Based Protein	Animal-Based Fat	Plant-Based Protein	Plant-Based Fat
	Median of Energy% (Minimum–Maximum) ^1^						
0	83.11 (80.80–93.06)	10.43 (4.92–11.00)	4.57 (1.94–6.20)	0.00 (0.00–0.84)	0.00 (0.00–1.36)	5.79 (2.19–6.44)	1.68 (0.66–1.91)
1	78.85 (77.09–80.78)	11.44 (11.00–11.81)	8.00 (6.20–9.45)	1.47 (0.85–1.94)	2.68 (1.37–4.14)	6.89 (6.44–7.25)	2.06 (1.91–2.26)
2	75.50 (73.95–77.07)	12.13 (11.81–12.45)	11.11 (9.48–12.47)	2.39 (1.94–2.77)	5.66 (4.14–7.09)	7.57 (7.25–7.83)	2.47 (2.26–2.67)
3	72.53 (71.14–73.95)	12.77 (12.45–13.04)	13.77 (12.47–14.83)	3.14 (2.77–3.49)	8.31 (7.09–9.41)	8.09 (7.83–8.32)	2.92 (2.67–3.17)
4	69.98 (68.61–71.14)	13.31 (13.04–13.55)	15.92 (14.84–17.03)	3.83 (3.49–4.17)	10.61 (9.42–11.66)	8.54 (8.32–8.80)	3.44 (3.17–3.70)
5	67.35 (66.01–68.60)	13.83 (13.55–14.12)	18.23 (17.04–19.36)	4.57 (4.17–4.97)	12.71 (11.66–13.83)	9.04 (8.80–9.32)	3.96 (3.70–4.27)
6	64.87 (63.51–66.01)	14.48 (14.12–14.85)	20.41 (19.36–21.60)	5.35 (4.97–5.81)	15.03 (13.84–16.05)	9.61 (9.32–9.91)	4.62 (4.27–5.00)
7	62.21 (60.69–63.50)	15.29 (14.86–15.76)	22.82 (21.61–24.04)	6.24 (5.81–6.77)	17.22 (16.06–18.66)	10.24 (9.91–10.58)	5.48 (5.00–6.08)
8	59.05 (57.36–60.69)	16.27 (15.76–16.88)	25.47 (24.04–27.10)	7.32 (6.77–7.95)	20.13 (18.66–21.76)	10.96 (10.58–11.42)	6.90 (6.08–7.84)
9	55.16 (52.49–57.34)	17.66 (16.88–18.74)	29.08 (27.10–31.37)	8.76 (7.96–9.95)	23.77 (21.76–26.54)	11.95 (11.42–12.62)	9.04 (7.84–10.67)
10	48.74 (17.49–52.48)	20.47 (18.74–35.75)	35.23 (31.37–64.22)	11.72 (9.96–28.77)	30.26 (26.54–63.15)	13.44 (12.62–35.02)	13.49 (10.68–37.60)

^1^ The overlap of values in parentheses is due to rounding the values to the second decimal place. Energy from diet total carbohydrate, total protein, total fat, animal protein, animal fat, plant protein, and plant fat is shown according the score assigned to the 11 groups after ranking the participants’ macronutrient intake, respectively.

**Table 2 nutrients-12-01307-t002:** General characteristics of Chinese adults according to the quartiles (Q) of the low-carbohydrate diet (LCD) scores, China Health and Nutrition Survey. ^1^

	Male					Female				
	Q1	Q2	Q3	Q4	*p* Value ^2^	Q1	Q2	Q3	Q4	*p* Value ^2^
N	669	735	656	683		824	794	786	774	
LCD Score (Median) ^3^	5	12	18	25		5	12	19	25	
LCD Range (Min–Max)	0–8	9–15	16–21	22–30		0–8	9–15	16–21	22–30	
Age (Years)	49.41 ± 0.57	49.07 ± 0.55	47.94 ± 0.58	48.45 ± 0.57	0.1171	48.88 ± 0.50	48.42 ± 0.52	48.46 ± 0.51	47.46 ± 0.53	0.0001
Age Groups (Years)										
18–29	73 (10.91)	72 (9.8)	79 (12.04)	81 (11.86)	0.5199	71 (8.62)	84 (10.58)	91 (11.58)	105 (13.57)	0.0570
30–39	107 (15.99)	137 (18.64)	117 (17.84)	121 (17.72)		126 (15.29)	142 (17.88)	140 (17.81)	146 (18.86)	
40–49	144 (21.52)	170 (23.13)	162 (24.7)	143 (20.94)		216 (26.21)	188 (23.68)	197 (25.06)	202 (26.1)	
50–59	170 (25.41)	181 (24.63)	158 (24.09)	185 (27.09)		214 (25.97)	197 (24.81)	190 (24.17)	168 (21.71)	
≥60	175 (26.16)	175 (23.81)	140 (21.34)	153 (22.4)		197 (23.91)	183 (23.05)	168 (21.37)	153 (19.77)	
Area of Residence ^4^										
Rural	481 (88.26)	436 (74.53)	338 (67.74)	312 (59.54)	<0.0001	692 (85.33)	547 (89.53)	431 (76.42)	356 (68.2)	<0.0001
Individual Net Income ^5^										
High	242 (44.9)	328 (56.65)	324 (65.72)	380 (72.66)	<0.0001	192 (32.32)	229 (41.49)	278 (53.56)	335 (62.5)	<0.0001
BMI (kg/m^2^)	22.59 ± 0.12	22.93 ± 0.12	23.31 ± 0.13	23.67 ± 0.13	<0.0001	23.12 ± 0.12	23.22 ± 0.12	23.18 ± 0.12	23.07 ± 0.12	0.7209
Weight Status ^6^										
Overweight	178 (26.61)	213 (28.98)	191 (29.12)	237 (34.7)	<0.0001	229 (27.79)	227 (28.59)	221 (28.12)	214 (27.65)	0.6930
Obese	33 (4.93)	53 (7.21)	62 (9.45)	74 (10.83)		76 (9.22)	73 (9.19)	66 (8.4)	61 (7.88)	
Alcohol Consumption ^7^										
Yes	398 (59.49)	431 (58.64)	421 (64.18)	474 (69.40)	<0.0001	54 (6.55)	65 (8.2)	68 (8.65)	109 (14.08)	<0.0001
Smoking Status ^8^										
Current Smokers	393 (58.74)	429 (58.45)	381 (58.08)	407 (59.59)	0.6072	16 (1.94)	21 (2.65)	17 (2.16)	18 (2.33)	0.5605
Physical Activity ^9^										
Light	193 (29.07%)	238 (33.06%)	252 (39.44%)	295 (43.96%)	<0.0001	132 (16.26%)	170 (21.6%)	151 (19.56%)	160 (21.16%)	<0.0001
Moderate	105 (15.81%)	109 (15.14%)	115 (18%)	147 (21.91%)		285 (35.1%)	325 (41.3%)	365 (47.28%)	399 (52.78%)	
Heavy	366 (55.12)	373 (51.81)	272 (42.57)	229 (34.13)		395 (48.65)	292 (37.1)	256 (33.16)	197 (26.06)	
Educational Level ^10^										
≥Technical School	330 (49.33)	412 (56.05)	371 (56.55)	371 (54.32)	<0.0001	292 (35.44)	328 (41.31)	344 (43.77)	355 (45.87)	<0.0001
Lipid Profile										
Total Cholesterol (mg/dL)	178.61 ± 1.35	185.21 ± 1.38	188.43 ± 1.44	193.09 ± 1.55	<0.0001	183.88 ± 1.33	189.64 ± 1.43	188.17 ± 1.34	190.74 ± 1.44	0.0019
Triglycerides (mg/dL)	129.27 ± 3.9	154.15 ± 5.5	159.85 ± 5.33	171.46 ± 6.45	<0.0001	132.03 ± 3.96	134.81 ± 3.5	125.88 ± 2.97	124.87 ± 3.47	0.0538
HDL-C (mg/dL)	55.7 ± 0.76	53.76 ± 0.6	52.74 ± 0.57	54.04 ± 0.75	0.0531	57.6 ± 0.53	57.6 ± 0.67	58.21 ± 0.52	58.49 ± 0.52	0.1939
LDL-Cl (mg/dL)	110.44 ± 1.67	111.59 ± 1.32	115.63 ± 1.37	116.99 ± 1.52	0.0003	112.19 ± 1.22	117.19 ± 1.33	116.39 ± 1.24	118.06 ± 1.32	0.0029

^1^ All statistical analyses accounted for the sampling design of the national surveys in the 2009 wave. Values are presented as the mean ± standard error, median, or *n* (%). ^2^
*p* values were calculated using the generalized linear model for continuous variables and the χ^2^ test for categorical variables. ^3^ The LCD scores range from a minimum of zero to a maximum of 30; a low LCD score indicates weaker adherence, whereas a high score indicates greater adherence and higher protein and fat intake. ^4^ Classified by “rural” and “urban” according to the questionnaire. ^5^ Classified into “low”, “middle”, “high” according the file from National Development and Reform Commission. ^6^ Determined by body mass index: underweight (<18.5 kg/m^2^), normal weight (≥18.5 kg/m^2^ and < 24 kg/m^2^), overweight (≥24 kg/m^2^ and <28 kg/m^2^), and obesity (≥28 kg/m^2^) according to the Guidelines for the Prevention and Control of Overweight and Obesity in Chinese Adults (2006). ^7^ “Yes” indicated drank alcohol over the past year. ^8^ “Past” indicated that the person has smoked over their lifetime but has not smoked recently, and “current” indicated that the person still smokes. ^9^ “Heavy” indicated the metabolic equivalent of task (MET) hours per week ≥ 3000, “moderate” indicated MET ≥ 600 and <3000, and “light” indicated MET < 600 according to the 2011 Compendium of Physical Activities. ^10^ Divided into four groups: “less than primary school”, “middle school”, “technical school”, and “college or university”.

**Table 3 nutrients-12-01307-t003:** Macronutrient intake and energy percent according to the LCD score based on gender, China Health and Nutrition Survey. ^1^

	Male					Female				
	Q1	Q2	Q3	Q4	*p* Value ^2^	Q1	Q2	Q3	Q4	*p* Value
N	669	735	656	683		824	794	786	774	
Food and Nutrient Intake										
Total Energy (kcal/d)										
	2348.78 ± 24.71	2339.9 ± 24.4	2425.81 ± 27.17	2370.44 ± 25.56	0.1879	1990.64 ± 20.69	1979.7 ± 20.09	2030.83 ± 20.8	1934.02 ± 19.56	0.2027
Carbohydrate (g/d)	406.71 ± 5.15	348.69 ± 3.85	327.59 ± 4.12	258.65 ± 3.46	<0.0001	335.28 ± 3.87	299.59 ± 3.5	270.45 ± 3.04	220.35 ± 2.72	<0.0001
Protein (g/d)	61.09 ± 0.78	66.93 ± 0.81	75.26 ± 0.99	83.03 ± 1.11	<0.0001	50.63 ± 0.60	57.87 ± 0.72	62.53 ± 0.75	70.56 ± 0.87	<0.0001
Fat (g/d)	20.85 ± 0.52	34.9 ± 0.57	52.65 ± 1.05	63.06 ± 0.97	<0.0001	16.82 ± 0.37	29.45 ± 0.49	42.55 ± 0.66	52.39 ± 0.82	<0.0001
Energy from Carbohydrates (%)										
Total	79.28 ± 0.15	70.72 ± 0.12	63.16 ± 0.18	53.31 ± 0.25	<0.0001	79.36 ± 0.14	70.79 ± 0.12	63.21 ± 0.14	53.84 ± 0.24	<0.0001
Energy from Protein (%)										
Total	11.93 ± 0.06	13.64 ± 0.07	14.64 ± 0.1	17.41 ± 0.13	<0.0001	12.00 ± 0.05	13.74 ± 0.07	14.72 ± 0.09	17.48 ± 0.11	<0.0001
Animal-Based	1.82 ± 0.06	4.15 ± 0.07	5.76 ± 0.09	8.86 ± 0.13	<0.0001	1.75 ± 0.05	4.05 ± 0.07	5.84 ± 0.08	8.7 ± 0.12	<0.0001
Plant-Based	10.1 ± 0.08	9.49 ± 0.08	8.89 ± 0.1	8.55 ± 0.11	<0.0001	10.25 ± 0.07	9.69 ± 0.08	8.89 ± 0.08	8.78 ± 0.11	<0.0001
Energy from Fat (%)										
Total	8.8 ± 0.16	15.64 ± 0.17	22.2 ± 0.25	29.28 ± 0.25	<0.0001	8.63 ± 0.14	15.46 ± 0.16	22.07 ± 0.2	28.68 ± 0.25	<0.0001
Animal Based	4.77 ± 0.17	11.12 ± 0.2	16.63 ± 0.29	22.54 ± 0.31	<0.0001	4.46 ± 0.15	10.51 ± 0.2	16.84 ± 0.25	22.06 ± 0.3	<0.0001
Plant Based	4.03 ± 0.09	4.52 ± 0.11	5.57 ± 0.17	6.74 ± 0.2	<0.0001	4.17 ± 0.08	4.95 ± 0.12	5.23 ± 0.13	6.62 ± 0.17	<0.0001

^1^ All statistical analyses accounted for the sampling design of the national surveys in the 2009 wave. Values are presented as the mean ± standard error. ^2^
*p* values were calculated using the generalized linear model for continuous variables.

**Table 4 nutrients-12-01307-t004:** Carbohydrate and fat intakes among Chinese adults according to the quartiles (Q) of the low-carbohydrate diet scores, China Health and Nutrition Survey. ^1^

	Male						Female					
	Total	Q1	Q2	Q3	Q4	*p* Value ^2^	Total	Q1	Q2	Q3	Q4	*p* Value ^2^
Compliance with AMDR ^3^ for Carbohydrates Recommended by the CDRI 2013 ^4^												
Low (<50%)	170 (6.20)	0 (0)	0 (0)	13 (1.98)	157 (22.99)	<0.0001	172 (5.41)	0 (0)	0 (0)	6 (0.76)	166 (21.45)	<0.0001
Moderate	956 (34.85)	0 (0)	20 (2.72)	410 (62.50)	526 (77.01)		1152 (36.25)	0 (0)	36 (4.53)	508 (64.63)	608 (78.55)	
High (>65%)	1617 (58.95)	669 (100)	715 (97.28)	233 (35.52)	0 (0)		1854 (58.34)	824 (100)	758 (95.47)	272 (34.61)	0 (0)	
Dietary Classification Based on the Amount of Carbohydrates Consumed ^5^												
Low (<40%)	37 (1.35)	0 (0)	0 (0)	2 (0.30)	35 (5.12)	<0.0001	34 (1.07)	0 (0)	0 (0)	2 (0.25)	32 (4.13)	<0.0001
Moderate	1089 (39.70)	0 (0)	20 (2.72)	421 (64.18)	648 (94.88)		1290 (40.59)	0 (0)	36 (4.53)	512 (65.14)	742 (95.87)	
High (>65%)	1617 (58.95)	669 (100)	715 (97.28)	233 (35.52)	0 (0)		1854 (58.34)	824 (100)	758 (95.47)	272 (34.61)	0 (0)	
Compliance with AMDR ^3^ for Fat Recommended by the CDRI 2013 ^4^												
Low (<20%)	1552 (56.58)	669 (100)	600 (81.63)	245 (37.35)	38 (5.56)	<0.0001	1827 (57.49)	824 (100)	656 (82.62)	294 (37.40)	53 (6.85)	<0.0001
Moderate	848 (30.92)	0 (0)	135 (18.37)	348 (53.05)	365 (53.44)		1007 (31.69)	0 (0)	138 (17.38)	435 (55.34)	434 (56.07)	
High (>30%)	343 (12.50)	0 (0)	0(0)	63 (9.60)	280 (41.00)		344 (10.82)	0 (0)	0 (0)	57 (7.25)	287 (37.08)	

^1^ All statistical analyses accounted for the sampling design of the national surveys in the 2009 wave. Values are presented as *n* (%). ^2^
*p* values were calculated by χ^2^ test for categorical variables. ^3^ AMDR = acceptable macronutrient distribution range. ^4^ CDRI 2013 = Chinese Dietary Reference Intakes (CDRIs) Handbook (2013); the AMDR for carbohydrates recommended by the CDRI is 50–65%. ^5^ Classification of the dietary carbohydrate level based on the previous modified criteria: low-carbohydrate diet, <40% of energy; moderate-carbohydrate diet, 40–65% energy; high-carbohydrate diet, >65% of energy.

**Table 5 nutrients-12-01307-t005:** Multivariate adjusted odds ratios (ORs) for dyslipidemia according to the quartiles (Q) of the low-carbohydrate diet (LCD) scores of Chinese adults, China Health and Nutrition Survey. ^1^

	Male					Female				
LCD Score	Q1	Q2	Q3	Q4	*p* Value for Trend ^2^	Q1	Q2	Q3	Q4	*p* Value for Trend ^2^
	OR (95% CI)					OR (95% CI)				
Hypercholesterolemia ^3^										
N (%)	55 (8.22)	79 (10.75)	86 (13.11)	103 (15.08)		85 (10.32)	109 (13.73)	106 (13.49)	110 (14.21)	
Adjusted Model ^4^	1.00	1.28	1.64	1.87	0.0017	1.00	1.46	1.36	1.42	0.1455
		(0.84–1.97)	(1.07–2.51)	(1.23–2.85)			(0.99–2.14)	(0.91–2.03)	(0.93–2.14)	
Hypertriglyceridemia ^5^										
N (%)	105 (15.7)	154 (20.95)	146 (22.26)	159 (23.28)		119 (14.44)	126 (15.87)	113 (14.38)	113 (14.6)	
Adjusted Model ^4^	1.00	1.24	1.25	1.47	0.0336	1.00	1.04	1.00	1.10	0.7060
		(0.89–1.72)	(0.89–1.76)	(1.04–2.06)			(0.73–1.49)	(0.69–1.45)	(0.72–1.60)	
Low High-Density Lipoprotein Cholesterol ^6^										
N (%)	88 (13.15)	117 (15.92)	123 (18.75)	121 (17.72)		74 (8.98)	62 (7.81)	52 (6.62)	56 (7.24)	
Adjusted Model ^4^	1.00	1.10	1.21	1.14	0.4427	1.00	0.87	0.48	0.87	0.1638
		(0.77–1.58)	(0.84–1.75)	(0.78–1.65)			(0.56–1.36)	(0.28–0.82)	(0.55–1.40)	

^1^ All statistical analyses accounted for the sampling design of the national surveys in the 2009 wave. Values are presented as *n* (%) or ORs (95%CIs). ^2^
*p* for trend values were calculated using the generalized linear model. ^3^ Hypercholesterolemia was defined as a total cholesterol level ≥ 6.22 mmol/L (240 mg/dL) on fasting blood glucose test, low-density lipoprotein cholesterol level ≥ 4.14 mmol/L (160 mg/dL). ^4^ Adjusted for age, educational level, body mass index, ethnicity, physical activity level, alcohol consumption, smoking status, and individual income. ^5^ Hypertriglyceridemia was defined as a triglyceride level ≥ 2.26 mmol/L (200 mg/dL). ^6^ A low high-density lipoprotein cholesterol (HDL-C) level was defined as a HDL-C level < 1.04 mmol/L (40 mg/dL).

**Table 6 nutrients-12-01307-t006:** Multivariate adjusted odds ratios (ORs) for dyslipidemia according to the quartiles (Q) of animal-based and plant-based low-carbohydrate diet (LCD) scores of Chinese adults, China Health and Nutrition Survey.

	Male					Female				
	Q1	Q2	Q3	Q4	*p* Value for Trend ^1^	Q1	Q2	Q3	Q4	*p* Value for Trend ^1^
	OR (95% CI)					OR (95% CI)				
Animal-Based LCD Score ^2^										
N	708	671	697	667		777	835	821	745	
Hypercholesterolemia ^3^										
N (%)	56 (7.91)	79 (11.77)	88 (12.63)	100 (14.99)		83 (10.68)	111 (13.29)	114 (13.89)	102 (13.69)	
Adjusted Model ^4^	1.00	1.66	1.8	2.15	0.0006	1.00	1.64	1.41	1.64	0.0651
		(1.08–2.54)	(1.17–2.77)	(1.41–3.29)			(1.10–2.44)	(0.93–2.13)	(1.06–2.55)	
Hypertriglyceridemia ^5^										
N (%)	127 (17.94)	133 (19.82)	144 (20.66)	160 (23.99)		116 (14.93)	126 (15.09)	123 (14.98)	106 (14.23)	
Adjusted Model ^4^	1.00	1.01	1.04	1.51	0.0156	1.00	1.09	1.13	0.94	0.8828
		(0.72–1.40)	(0.74–1.45)	(1.09–2.10)			(0.76–1.56)	(0.77–1.64)	(0.63–1.41)	
Low High-Density Lipoprotein Cholesterol ^6^										
N (%)	46 (6.50)	64 (9.54)	73 (10.47)	78 (11.69)		70 (9.01)	92 (11.02)	96 (11.69)	89 (11.95)	
Adjusted Model ^4^	1.00	0.97	1.08	1.06	0.6618	1.00	0.77	0.79	0.63	0.1101
		(0.68–1.39)	(0.75–1.54)	(0.73–1.52)			(0.49–1.22)	(0.49–1.28)	(0.37–1.08)	
Plant-Based LCD Score ^2^										
N	736	617	634	756		776	899	767	736	
Hypercholesterolemia ^3^										
N (%)	95 (12.91)	69 (11.18)	68 (10.73)	91 (12.04)		96 (12.37)	125 (13.9)	83 (10.82)	106 (14.40)	
Adjusted Model ^4^	1.00	0.81	0.72	0.82	0.2929	1.00	1.15	0.81	0.98	0.5608
		(0.55–1.20)	(0.48–1.07)	(0.56–1.18)			(0.80–1.67)	(0.55–1.21)	(0.67–1.46)	
Hypertriglyceridemia ^5^										
N (%)	145 (19.70)	122 (19.77)	141 (22.24)	156 (20.63)		102 (13.14)	132 (14.68)	125 (16.30)	112 (15.22)	
Adjusted Model ^4^	1.00	0.86	0.95	0.77	0.1386	1.00	1.1	1.23	1.35	0.0951
		(0.62–1.20)	(0.69–1.32)	(0.56–1.06)			(0.76–1.60)	(0.85–1.79)	(0.92–1.97)	
Low High-Density Lipoprotein Cholesterol ^6^										
N (%)	85 (11.55)	57 (9.24)	46 (7.26)	73 (9.66)		84 (10.82)	107 (11.90)	66 (8.60)	90 (12.23)	
Adjusted Model ^4^	1.00	0.76	0.88	0.92	0.8642	1.00	1.31	1.1	1.21	0.6380
		(0.52–1.09)	(0.62–1.25)	(0.66–1.30)			(0.82–2.12)	(0.67–1.80)	(0.73–2.00)	

^1^*p* for trend values were calculated using the generalized linear model. ^2^ The LCD scores range from a minimum of zero to a maximum of 30; a low LCD score indicates weaker adherence, whereas a high score indicates greater adherence and higher protein and fat intakes. ^3^ Hypercholesterolemia was defined as a total cholesterol level ≥ 6.22 mmol/L (240 mg/dL) on a fasting blood glucose test, low-density lipoprotein cholesterol level ≥ 4.14 mmol/L (160 mg/dL). ^4^ Adjusted for age, educational level, body mass index, ethnicity, physical activity level, alcohol consumption, smoking status, and individual income. ^5^ Hypertriglyceridemia was defined as a triglyceride level ≥ 2.26 mmol/L (200 mg/dL). ^6^ A low high-density lipoprotein cholesterol (HDL-C) level was defined as a HDL-C level < 1.04 mmol/L (40 mg/dL).

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
