# Peer review of "Association between Three Low-Carbohydrate Diet Scores and Lipid Metabolism among Chinese Adults"

_nutrients, 2020, doi:10.3390/nu12051307_

Round 1

Reviewer 1 Report

In this paper, the authors investigated the blood lipid levels of 5921 Chinese adults aged >18 years using data from the China Health and Nutrition Survey 2009. Diet information was collected through 3  day 24-h recalls by trained professionals. Using logistic regression models, the authors conclude that a higher LCD score, indicating lower carbohydrate intake and higher fat intake, especially animal-based fat, is significantly associated with higher odds of  hypercholesterolemia and hypertriglyceridemia in Chinese male. 

This is a very interesting study and would appeal to a good audience in the field of nutrition and metabolism. The authors did a reasonable good job with presenting the results. However, a few concerns:

1) Since physical activity has a big impact on development of dyslipidemia, please elaborate a little more on the level of physical activities in the male with different LCD scores. Is there a pattern for higher LCD score population to have less physical activity? Discuss the results accordingly

2) Alcohol consumption also has a big impact on the development of dislipidemia, please discuss the data for alcohol consumption across different LCD scores. 

3) Table 4-6  are not labeled right, please make the correct label.

Author Response

We appreciate the opportunity to revise our manuscript. We carefully considered your opinions and revised the manuscript based on your comments. Detailed explanations regarding the changes we have made are described in the following responses.

COMMENT 1. Since physical activity has a big impact on development of dyslipidemia, please elaborate a little more on the level of physical activities in the male with different LCD scores. Is there a pattern for higher LCD score population to have less physical activity? Discuss the results accordingly.

COMMENT 2. Alcohol consumption also has a big impact on the development of dyslipidemia, please discuss the data for alcohol consumption across different LCD scores.

RESPONSE: In response to the reviewer’s comments 1 and 2, we have discussed further.

[page 10, lines 286-293]

Furthermore, in the present study, it was also found that participants with higher LCD scores lived in urban areas, had high income, consumed alcohol, performed light physical activity, and had higher educational levels. Among these, the PAL and drinking status have been widely studied and confirmed as important factors on lipid metabolism[4,21,22]. Due to their inescapable impact, in present study, we adjusted PAL and drinking status as covariables in the multivariate model to minimize their impact, improve the validity of research results and focus on the analyses of impact of diet on lipid metabolism. In the follow-up study, we will consider PAL and drinking status as exposure factors to study their impact on lipid metabolism.

COMMENT 3. Table 4-6 are not labeled right, please make the correct label.

RESPONSE: Thanks for your comment. Per your comment, we have revised the label.

[page 7, lines 225-226]

Table 4. Carbohydrate and fat intake levels of Chinese adults according to quartile (Q) of the low-carbohydrate diet scores, China Health and Nutrition Survey1

[page 8, lines 246-247]

Table 5. Multivariate adjusted odds ratios (ORs) for dyslipidemia according to the quartile (Q) of the low-carbohydrate diet (LCD) scores of Chinese adults, China Health and Nutrition Survey1

[page 8, lines 268-269]

Table 6. Multivariate adjusted odds ratios (ORs) for dyslipidemia according to the quartile (Q) of animal-based and plant-based low-carbohydrate diet (LCD) scores of Chinese adults, China Health and Nutrition Survey

Reviewer 2 Report

page 14 line 452 all words in the title of journal shuld be from big letters

page 14 line 461 all words in the title of journal shuld be from big letters

Author Response

We appreciate the opportunity to revise our manuscript. We carefully considered your opinions and revised the manuscript based on your comments. Detailed explanations regarding the changes we have made are described in the following responses.

COMMENT 1.

page 14 line 452 all words in the title of journal should be from big letters

COMMENT 2.

page 14 line 461 all words in the title of journal should be from big letters

RESPONSE: We thank your comments and we have changed all words in the title of journals to capital letters.

[page 14, lines 470-471]

Yokoyama, Y.; Levin, S.M. Association between Plant-based Diets and Plasma Lipids: A Systematic Review and Meta-Analysis. NUTR REV 2017, 75, 683-698, doi:10.1093/nutrit/nux030.

[page 14, lines 478-480]

Knopp, R.H.; Paramsothy, P. Gender Differences in Lipoprotein Metabolism and Dietary Response: Basis in Hormonal Differences and Implications for Cardiovascular Disease. CURR ATHEROSCLER REP 2005, 7, 472-479, doi:10.1007/s11883-005-0065-6.

Reviewer 3 Report

Tan et al. provide a large study conducted on Chinese adults with the aim to assess the association between three low-carb diet scores and lipid metabolism.

I read with interest this study, which has many strenghts (well conducted study, large population, clinically relevant topic).

I do however have some suggestions:

  • in line 53, please reference the previous studies
  • in line 60, please spell in full the acronym (LCD)
  • Please provide in the statistical analyses section the variables that have been adjusted for in the logistic regression analysis
  • in line 224, the number of the table should be 4, in line 245 the table should be 5 and in line 267 the table should be 6
  • Information on the duration of follow-up should be included in the "materials and methods" section

Author Response

We appreciate the opportunity to revise our manuscript. We carefully considered your opinions and revised the manuscript based on your comments. Detailed explanations regarding the changes we have made are described in the following responses.

COMMENT 1.

in line 53, please reference the previous studies

RESPONSE: Thanks for your comment. We have added the reference.

[page 2, lines 50-53]

Among these influencing factors, obesity (specifically abdominal obesity), alcohol consumption, and high-fat diets (specifically high saturated and trans-fatty acid intakes) are well-known risk factors of dyslipidemia, as confirmed by several previous studies[3-5].

COMMENT 2.

in line 60, please spell in full the acronym (LCD)

RESPONSE: In response to the reviewer’s comments, we have spelt in full name the LCD.

[page 2, lines 60-61]

The concept and calculation method of the low carbohydrate diet (LCD) score have been introduced in detail previously [9,11,12].

COMMENT 3.

Please provide in the statistical analyses section the variables that have been adjusted for in the logistic regression analysis

RESPONSE: Thanks for your comment, we have provided the information for covariates in the logistic regression model.

[page 5, lines 167-170]

After adjusting for potential confounding variables (including age, educational level, body mass index, ethnicity, physical activity level, alcohol consumption, smoking status, and individual income), logistic regression models were used to calculate the odds ratios (ORs) of dyslipidemia and their 95% confidence intervals (95% CIs).

COMMENT 4.

in line 224, the number of the table should be 4, in line 245 the table should be 5 and in line 267 the table should be 6

RESPONSE: We have revised the number of the Table 4-6.

[page 7, lines 225-226]

Table 4. Carbohydrate and fat intake levels of Chinese adults according to quartile (Q) of the low-carbohydrate diet scores, China Health and Nutrition Survey1

[page 8, lines 246-247]

Table 5. Multivariate adjusted odds ratios (ORs) for dyslipidemia according to the quartile (Q) of the low-carbohydrate diet (LCD) scores of Chinese adults, China Health and Nutrition Survey1

[page 8, lines 268-269]

Table 6. Multivariate adjusted odds ratios (ORs) for dyslipidemia according to the quartile (Q) of animal-based and plant-based low-carbohydrate diet (LCD) scores of Chinese adults, China Health and Nutrition Survey

COMMENT 5.

Information on the duration of follow-up should be included in the "materials and methods" section

RESPONSE: We thank your comment. Although CHNS is a large-scale national representative study, current study was conducted a cross-sectional study design using data from 2009 year of CHNS. Therefore, there was no follow-up information in our study.

Round 2

Reviewer 2 Report

I accept this paper.